# Intraluminal Impact of Food: New Insights from MRI

**DOI:** 10.3390/nu11051147

**Published:** 2019-05-23

**Authors:** Robin Spiller, Luca Marciani

**Affiliations:** Nottingham Digestive Diseases Centre and NIHR Nottingham Biomedical Research Centre, University of Nottingham, Nottingham NG7 2UH, UK; Luca.Marciani@nottingham.ac.uk

**Keywords:** magnetic resonance imaging, stomach, small bowel, colon, food

## Abstract

Understanding how the gut responds to food has always been limited by the available investigatory techniques. Previous methods involving intubation and aspiration are largely limited to liquid-only meals. The aim of this review is to describe how MRI has allowed analysis of the processing of complex multiphase meals. This has demonstrated the role of physical factors such as viscosity, fat and fibre content in controlling gastric secretion and motility. It has also allowed the description of changes induced in small bowel water content and the role of osmotic effects of poorly absorbed carbohydrates such as fructose, sorbitol and mannitol. Intestinal secretions can be shown to be stimulated by a range of fruit and vegetables and the effect of this on colonic water content can also be measured. This has been used to demonstrate the mode of action of commonly used laxatives including bran and psyllium. The wealth of data which can be obtained together with its non-invasive nature and safety makes the technique ideal for the serial evaluation of the impact of different nutrients and drugs in both health and disease.

## 1. Introduction

Imaging the gut using MRI has proven to be a powerful new tool in understanding how the gut responds to the complex, multinutrient, multiphasic meals we typically consume. Previous techniques which relied on intubation and aspiration to define what happens after meal ingestion were essentially limited to low viscosity fluids which could be aspirated up narrow oral/nasal tubes, whose size was limited by what could be reasonably tolerated by volunteers. MRI by contrast can image and quantify the movement of liquid/solid mixtures both within the stomach and the small and large bowel, irrespective of their viscosity. The earliest MRI studies were focused on the stomach which provided clear easily analysable images. This selected review will use studies mostly from our own laboratory but also from others to describe how food alters intraluminal content in the stomach, small and large intestine.

## 2. Magnetic Resonance Imaging

Magnetic resonance imaging (MRI) is a medical imaging technique that can acquire detailed images of the body and of foods and beverages inside the gastrointestinal tract non-invasively. 

The method is based on the use of radiofrequency and therefore does not use harmful ionizing radiation. As such, it is ideal for studies of gastrointestinal function and its response to feeding. Briefly, when the human body (or a food or beverage) is placed inside a strong magnetic field like that of an MRI scanner, it is possible to transmit and receive radiofrequency signals to and from the hydrogen protons. The signal depends on the magnetic field hence it is possible to encode spatial information using three orthogonal magnetic field gradients. The encoded signal can be later post processed thereby reconstructing the MRI image. The signal also depends on the physicochemical characteristics of the tissue or material being imaged. This dependence is characterized by two principal time constants, one reflecting the recovery of the longitudinal magnetization after a radiofrequency excitation, and is termed T1 recovery, and the other constant reflects the gradual decay of the magnetization in the transverse plane and is called T2 decay. 

The MRI images can be weighted to contain different contributions from T1 and T2, which is at the heart of the richness of contrast in the MRI images. Additional information can be captured by sensitizing the images, for example, to flow and diffusion.

## 3. Gastric Processing of Complex Foods

Imaging of intra-gastric contents demonstrates a wealth of detail previously unrecognized and unmeasurable when studies were limited to measurement of merely the rate of gastric emptying using scintigraphy (Figure 1). When food is ingested, chewing and mixing with saliva play a crucial role in modifying its consistency to form a bolus, whose physical properties allows easy passage past the larynx and down the oesophagus into the stomach [1]. The successive food boluses are accommodated within the stomach by reflex fundal relaxation. Soon after ingestion, antral contractions, starting mid-way down the lesser curvature, elute material from the outside of the solid component of the meal into the antrum and thence through the pylorus. MRI sequences sensitive to flow have been used to demonstrate the backward and forward movements induced by a combination of antral contractions and pyloric closure, which results in both retropulsion of contents back into the antrum and propulsion of a fine stream through the pylorus [2]. The retropulsive movements we observed were most marked after mixed nutrient fat/protein/carbohydrate meals compared to 5% or 10% glucose. The shear forces generated by antral grinding are surprisingly small compared to those in the mouth, and when assessed from the force required to break agar gel beads were found to be <0.65 newton [3]. 

## 4. Gastric Sieving 

The antral contractions are typically propulsive as they commence in the mid stomach but as they approach the pylorus change to a more systolic pattern with pyloric closure occurring before all the chyme is expelled into the duodenum. This effectively restricts gastric emptying to particles <2 mm in diameter and creates shearing forces as the remaining material is forced backwards away from the contracting pylorus. The resulting forward and backward flows can be detected using flow sensitive MRI sequences as mentioned previously. The net effect is the earlier emptying of the fluid component of mixed liquid/solid meals, which make up most of what normal humans ingest. MRI images can clearly show the heterogeneity of food immediately after food ingestion, which varies according to the meal type (Figure 1). A rice pudding meal typically separates into an uppermost fluid layer above a dependent solid component, whereas a roast chicken and vegetable meal produces a very heterogeneous intragastric distribution of solid and liquid. Other more homogenous meals like porridge, soup or bread produce a uniform intra-gastric mass from which no separation of the liquid and solid phase is possible. When we compared an equicaloric rice meal with wholemeal bread, we found that while the rice pudding meal separated into a solid and liquid phase, the wholemeal bread formed a rather homogeneous mass with no clear separation of fluid and solid. While the rice meal allowed the stomach to sieve the meal and empty the fluid faster than the solid, sieving was not possible with the more homogeneous bread meal and as a consequence the gastric volume fell more slowly. In contrast, once the wheat entered the small intestine it was rapidly absorbed and the small bowel water content (SBWC) was consistently lower than after the rice meal postprandially, suggesting than any sensation of bloating after a bread meal most probably comes from gastric distension [4]. Other groups have also used MRI to document gastric sieving and its prevention by homogenization with both solid [6] and liquid [7] meals.

In Figure 2 it is shown how a typical solid meal taken with water drink (D–E) fills the stomach with heterogeneous content from which the fluid can be preferentially emptied causing a faster fall in total gastric volume and hence feeling of fullness. We compared this with the same meal blenderised to create a homogeneous soup (images A–C) and showed that the volume of the stomach fell more slowly with the soup and this was associated with more prolonged satiety [5].

## 5. Effect of Gastric Secretion on Intra-Gastric Dilution of the Meal

Regardless of the initial meal viscosity, the addition of salivary and then gastric secretions mean that the meal is rapidly diluted. This can be assessed by performing serial imaging at different times after proton excitation (See Figure 3). The time constant of the decay of image intensity, known as T2, which reflects the interaction of water molecules with their environment can be used as a measure of intra-gastric dilution and from in vivo calibration this can also be used as a measure of viscosity of meals of known composition. Using T2 we could show that intra-gastric chyme viscosity rapidly falls, with an immediate reduction of around 10% due to saliva. Gastric secretions continue this process so that by 40 min it is diluted around 50% with an associated 10-fold fall in viscosity compared to the meal viscosity prior to ingestion. The homogeneity of intra-gastric dilution is strongly dependent on meal viscosity. Thus, for a highly viscous (just pourable viscosity 17 Pa·s) locust bean gum meal, a viscosity gradient persists, with the viscosity as assessed by T2 in the centre of the gastric lumen remaining virtually unchanged from the original value, even after 48 min. This may be important, because it will allow salivary lipase to continue to act within a swallowed bolus, even though its pH optimum is several log units higher than that of gastric acid. 

MRI using a gadolinium contrast-containing fluid allow the detection of gastric acid secretion, which is added to the meal, increasing the total volume in the first 30 min postprandially and creating a peripheral rim of gastric acid. The total gastric secretion can be quantified from the dilution of gadolinium contrast [8] confirming previous findings by others [9,10]. The more viscous meals like porridge show prolonged retention with elution of the periphery which leaves the interior of the food bolus largely intact.

The sensation of fullness after meal ingestion correlates strongly with gastric volume (Figure 4) and is greater at all time points in high viscosity compared with low viscosity meals [11]. This was also shown by others using meals with different macronutrient composition [12] showing the importance of gastric volume and viscosity in influencing satiety [13,14]. Whether this reflects the greater force of gastric contractions required to process the more viscous meal or, more likely, the different perception and longer time taken during swallowing [15], remains to be more clearly defined. Interestingly when agar beads are included in the meal, harder beads which require greater forces to fracture them induce a greater sense of fullness suggesting that antral wall tension may also influence the sense of fullness [11].

The relation between gastric distension and fullness sensation is important since it means that delaying emptying by modifying critical physical factors of food such as meal viscosity and homogeneity, as was shown in the soup studies referred to above [5], could be used as adjuncts to weight reducing programmes.

The effect of water ingested after a meal has also been extensively studied by MRI and confirms earlier studies in showing that water easily passes around gastric contents, particularly solid components, and empties into the small bowel as quickly as if the stomach were empty. It appears to preferentially pass down the lesser curvature, where the longitudinal gastric folds form what has been termed the “magenstrasse” (stomach road) [16]. 

## 6. Effect of Gluten on Intra-Gastric Processing of Bread

There has been much interest in the possible adverse effects of gluten in functional gastrointestinal disease and many patients complain that bread induces bloating. We have investigated the effect of gluten on the upper intestinal processing of 308 g bread meals comparing 2 days of feeding gluten-free bread (GFB) with the effect of bread with normal gluten content and a bread supplemented with 3% gluten. The gastric images showed the added gluten produced a less homogeneous intra-gastric appearance, but this was not associated with any difference in gastric emptying and all meals showed the same pronounced drop in small bowel water postprandially [17]. Unexpectedly, feeding GFB for 2 days significantly increased the fasting transverse colonic volumes by 36% compared to normal bread, possibly reflecting the effect of the fermentation of residues from the tapioca and potato starches used to replace wheat in GFB. Later studies have also showed a rise in colonic volumes on a low FODMAP (Fermentable Oligo-, Di-saccharide And Polyhydric alcohols) diet, which includes the substitution of normal with gluten-free bread [18]. Feedback from the colon via short chain fatty acid production may well be important in appetite control and several studies are attempting to exploit this in the management of obesity.

## 7. Effect of Emulsification of Fat on Intra-Gastric Distribution and Gastric Emptying

One key factor determining gastric emptying is duodenal sensing of nutrients and this depends on when nutrients first leave the stomach. This was clearly demonstrated using an unemulsified oil–water mixture [19]. When subjects lay with their left side uppermost, the fat remained in the stomach floating in the fundus while the water phase emptied rapidly. However, if the subjects lay with their right side uppermost, the fat entered the duodenum much earlier and gastric emptying was much delayed and satiety prolonged. Most fat is of course consumed as an emulsion, but gastric acidity means that many emulsions “crack” on entering the stomach, leaving a layer of the less dense but more calorific fat floating above the water phase. In the upright position, the water phase which contains little fat can leave the stomach rapidly since it does not excite duodenal receptors which detect fatty acids, releasing cholecystokinin (CCK) and thereby delaying gastric emptying. MRI shows that making emulsions more stable using emulsifiers can prevent cracking (Figure 5A) and result in much delayed gastric emptying (Figure 5B) [20]. Interestingly, such stable emulsions appear more efficient in releasing CCK, which may be why they are more satiating. 

During normal meal consumption it is common to consume courses, such as pates or mousses with a high fat content separate from the rest of the meal. Under such circumstances MRI has been able to demonstrate that the intra-gastric behaviour of fat depends on the order in which fat is consumed [21]. Taking it first results in earlier filling of the antrum and more layering as it meets the resting gastric acid, relatively undiluted by the non-fat component of the subsequent meal. When consumed second the fat mixed with the larger non-fat component whose buffering of gastric acid may account for reduced cracking of the emulsion and less layering. The same group, which assessed both intra-gastric secretions as well as fat, has recently confirmed that emulsion stability was a key determinant of fat layering, and also showed that the more stable emulsion increases the accumulation of gastric secretion in the antrum [22].

## 8. Effect of Simple Sugars, Polyhydric Alcohols and Osmotic Laxatives on Small Bowel Contents

While intestinal perfusion studies have shown that glucose is rapidly absorbed by a 30 cm perfused jejunal segment it has not been possible to quantitatively translate this into the effect in an intact human being until the advent of MRI studies. Isotonic glucose can be shown to be absorbed very rapidly so that 40 min after ingesting 350 mL of 5% glucose, 250 mL has left the stomach, but the small bowel contains only 50 mL of water implying that 200 mL has been absorbed over this time (Figure 6) [23].

By contrast at the same time after 250ml of mannitol has left the stomach the small bowel contains nearly 400ml indicating a net secretion of 150ml. The effect of mannitol is most likely due to its osmotic activity and very similar effects are known to occur after ingestion of other osmotically active, unabsorbable small molecules. 

Lactose intolerance has been studied in detail using intestinal intubation and aspiration showing a 10-fold dilution of the non-absorbable marker polyethylene glycol (PEG) after ingestion of 50 g lactose in 180 mL of water. This implies that around 1600 mL of water had entered the small bowel, most likely driven by the low sodium concentration and hypertonicity of the ingested solution [24]. Previous perfusion studies indicate that the luminal Na+ concentration strongly influences Na+ and water flow with net secretion at concentrations <90 mEq/L in mixed saline nutrient solutions [25] 

Lactulose, a nonhydrolyzable disaccharide, similarly induces small bowel secretion with an increase of around 150 mL of small bowel water 1 hour after ingestion as shown by MRI [26]. Moviprep^®^, an osmotic laxative composed of a PEG solution containing sodium chloride, sulphate and ascorbate caused a rapid increase in SBWC, from a baseline of 80–100 mL raising to peak around 650 mL after 1 L of Moviprep^®^. [27]. There was also a rapid transfer of fluid to the colon, which showed as a much brighter image with an associated rise in T2. The effect of mannitol was much less apparent and at the dose used the colonic images changed surprisingly little, indicating that most of the mannitol was fermented and the resulting short chain fatty acids (SCFAs) absorbed. Small bowel water content has also been shown to be increased markedly by grapefruit juice [28]. This is of interest since grapefruit juice is known to have important effects on the bioavailability of a number of drugs. Initially it was thought to be due to inhibition of key drug metabolizing enzymes such as intestinal CYP3A enzymes and by inhibiting transporters such as the organic anion transporting polypeptide (OATP). However, an additional factor could be that grapefruit juice increases the small bowel water and thus aids drug absorption. 

## 9. Effect of FODMAPs on the Small Bowel and Colon

Fructose is also an osmotically active poorly absorbed monomer, which behaves somewhat like mannitol and lactulose increasing small bowel water. It is an important dietary component and in some parts of the world daily intake is substantial, being 12.7 g in a typical Australian diet [29]. When 50 g of fructose is given in 500 mL of water, SBWC rises to over 400 mL [30]. Low FODMAP diets emphasize the concept of fructose in excess of glucose since animal studies show that glucose can enhance fructose absorption. Our MRI study also compared 50 g of fructose with 50 g of glucose and 50 g glucose plus fructose and clearly showed that adding glucose slowed gastric emptying and reduced the increase in SBWC (Figure 7A). This also reduced the increase in small bowel diameters (Figure 7B).

Patients with the irritable bowel syndrome often develop symptoms after consuming fructose-containing foods, but this appears not to be due to abnormally increased secretion or worse malabsorption since when challenged with either fructose or inulin they showed similar increases in both SBWC and colonic volumes as were seen with healthy controls [31]. 

## 10. Effect of Fibre on Small Bowel and Colonic Water Content

Dietary fibre comprises a range of materials derived from plant-supporting structures and storage polysaccharide, which are divided into soluble and insoluble forms. These are widely used to treat constipation. MRI has enabled us to understand the underlying mechanisms. Psyllium appears to trap water in the small bowel by increasing viscosity thereby impairing convection and reducing access of water to the intestinal mucosa where absorption takes place. Therapeutic doses of 3.5 g and 7 g three times daily with food lead to increased postprandial SBWC, which rises steadily during the day until meal two, a large (1000 kcal) meal stimulates ileal emptying, reducing SBWC back towards the baseline (Figure 8) [32].

Our earlier studies with bran [23], and more recent studies comparing bran directly with psyllium and prickly pear extract [33], show a similar increase in SBWC, though in this case the increase appears to be more likely due to stimulation of secretion since bran shows little water trapping [34].

The residual watery ileal contents are emptied into the ascending colon after a large meal which is associated with an increase in fluidity of the ascending colon contents, shown by an increase in T1 (Figure 9A,B). T1 is a time constant reflecting the recovery of longitudinal magnetization after excitation by radiofrequency energy pulses and increases as the colonic contents become more watery. During the study day, the T1 was significantly higher after both 3.5 g and 7 g psyllium being 631 ± 37 and 779 ± 48 ms, respectively, compared to the maltodextrin control at 447 ± 37 ms. We validated the use of T1 measurements as an assessment of colonic water content by showing that T1 of descending colon contents were well correlated with the percentage water of stool passed subsequently (Pearson’s r = 0.65; Figure 9C) [32].

## 11. Effect of Phytochemicals

During evolution there has been a constant battle between plants and herbivores or insects as plants develop toxins to inhibit grazing by animals or insects. Humans, by developing cooking, have been able to exploit these substances to enhance food attractiveness and add flavours. However, some of these toxins exert a laxative effect, often mediated by prostaglandin secretion since, like the laxative effect of sennosides [35] and Bisacodyl^®^ [36], they can be blocked by the cyclooxygenase inhibitor, indomethacin. 

Rhubarb creates substantial amounts of the mild cytotoxin, rhein, which is a powerful laxative widely used in Chinese herbal medicines. We recently showed that the laxative properties of rhubarb are associated with a marked increase in postprandial small bowel water, which is followed by an increase in ascending colon T1 [37]. We also found that lettuce similarly increased small bowel water, possibly due to the latex-like bitter lactucins they contain, which inhibit insect attacks. This may explain why ileostomists report that both lettuce and rhubarb increase ileostomy output [38].

Other plant defences are used by kiwifruit, which contains needle-like calcium phosphate crystals, rafides, as well as the cysteine protease, actinidin. Clinical trials have shown that regular kiwifruit can increase stool frequency and soften stool in healthy subjects, and a kiwifruit extract has been developed to treat constipation successfully [39,40]. Two kiwifruits taken twice daily demonstrate an immediate increase in small bowel water following consumption, which increases steadily throughout the day only falling after a second large meal which stimulates ileal emptying into the ascending colon. This was associated with a rise in ascending colon T1, indicating increased water content caused by increased inflow of ileal fluid [41].

## 12. Effect of Physical Form of Food

It has long been known that the physical form of food has an important effect on satiety with whole foods [42] and those with high fibre content resulting in more prolonged satiety. A recent MRI study has made it possible to monitor gastric, small and large intestine contents postprandially. This showed that whole apples produced a more heterogeneous intra-gastric bolus, which emptied slower and was associated with more prolonged fullness and reduced hunger. The satiating effect lasted several hours after the stomach was emptied, indicating the importance of intestinal effects. SBWC was greater with the solid apple, as was the colonic volumes, suggesting undigested material was entering the colon and products of microbiota-induced fermentation may play a part in satiety (Krishnasamy et al., manuscript in preparation). A greater understanding of how physical factors influence the gut may be useful in the future in guiding dieticians’ advice to patients with food intolerances, since slowing delivery by taking less processed foods may well improve tolerance and simplify exclusion diets.

## 13. Limitations and How These Might Be Overcome

A major impediment to using MRI is its expense. This can in part be overcome with greater knowledge of the time course of events. By allowing studies to make less frequent measurements at critical time points, the costs can be reduced many fold. Another limitation is that what is measured is a net effect and cannot separate easily changes in secretion versus changes in absorption. This can in part be overcome by adding contrast labels such a gadolinium and measuring its dilution as has been done to measure gastric secretion. However, this could only be used in the small intestine if contrast was given via a naso-intestinal tube, which would introduce artefact and detract from the otherwise non-invasive features of the technique.

Although the volumes of organs such as the stomach and colon are often what gives rise to symptoms, mechanistically we are often interested in flow rather than just intestinal volumes. This requires a more accurate measurement of transit to allow calculation of flow from the formula Flow = Volume/Transit Time. Our current five marker pill technique is far from ideal. Firstly, the pills are large and may not behave as liquids do, and secondly, it is well-known that the more pills one uses the more reproducible the marker technique becomes, with more than 10 being ideal [43]. The obvious need is for much smaller pills. The development of indigestible MRI-visible mini-capsules, which move down the gut at the same speed as chyme, is currently an active area of research. 

## 14. Conclusions and Outlook

MRI has opened up some exciting possibilities allowing non-invasive assessment of intra-gastric processing of complex foods and intestinal fluid distribution in response to both food and drugs. Assessment of intra-gastric viscosity and flow, intestinal water content and regional colonic volumes is possible at the same session using different MRI sequences. Water content in the colon can be measured using the T1 parameter, which allows assessment of response to treatments aimed at altering stool consistency. Whilst some of the methods are now established, there are many additional developments that will expand the application of MRI to the understanding of the intraluminal impact of food. The ability of MRI to selectively image the fat components has only started to be explored but could be used to examine fat malabsorption. Magnetic resonance spectroscopy (MRS) has also been widely applied in other fields to look at metabolites, but the applications to food and beverages in the body has been thus far limited. The techniques can be used to assess the mode of action of novel food materials and products to facilitate decisions on industrial development using relatively small numbers, before committing to large-scale, very expensive human trials. Another aspect under constant evolution is increasing imaging speed to counteract respiratory and organ motion, and also automation of data processing which can still be a burden. Some applications may well move MRI from its current purely research role into clinical practice. Imaging the response of constipated patients to an osmotic laxative clearly distinguishes those with an inert hypomotile colon from those with a hyperactive irritable bowel and may well guide treatment in the future [44,45]. Whether individual differences in responses to dietary items might also guide diet therapy remains to be determined.

## Figures and Tables

**Figure 1 nutrients-11-01147-f001:**
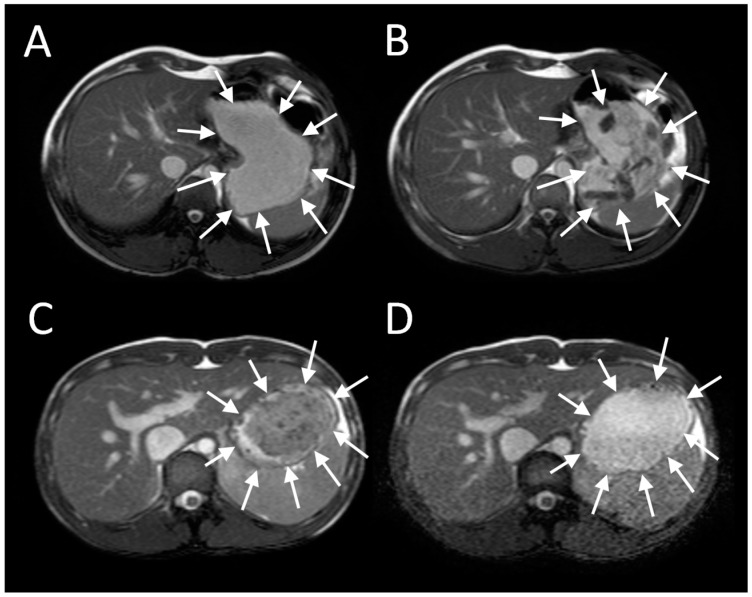
Gastric MRI images of different mixed solid/liquid meals: (**A**) Soup, (**B**) roasted chicken and vegetables, (**C**) whole wheat bread and (**D**) rice pudding. Reproduced from [4], Springer Nature 2013, under Creative Commons CC-BY license and from [5], Oxford University Press 2012, with permission.

**Figure 2 nutrients-11-01147-f002:**
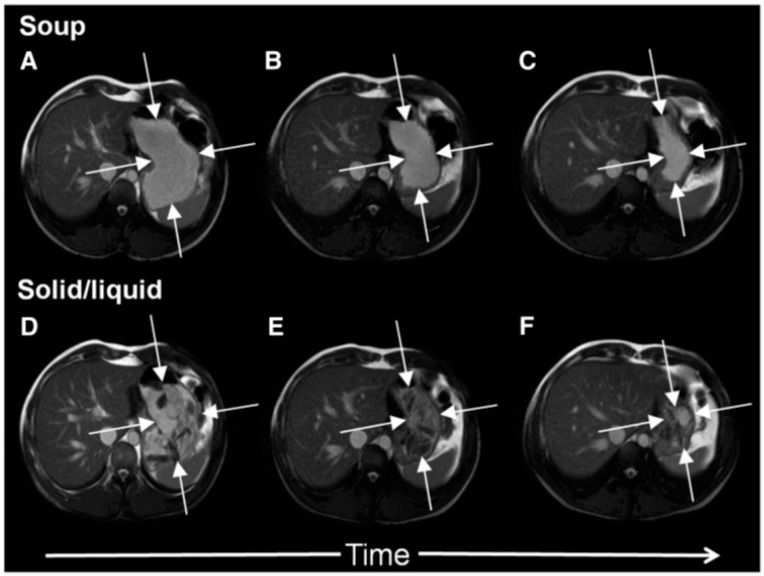
Cross-sectional MRI images showing a stomach at 0, 60 and 120 min after ingesting either roast chicken and vegetables (panels D, E and F respectively) or the same meal blenderised to create a homogeneous soup (panels A, B and C respectively). The solid meal taken with water appears heterogeneous, fluid can be preferentially emptied reducing gastric volume and hence the feeling of fullness. The same meal when blenderised appears homogeneous, gastric volumes fall more slowly with time which is associated with more prolonged satiety. Reproduced from [5], Oxford University Press 2012, with permission.

**Figure 3 nutrients-11-01147-f003:**
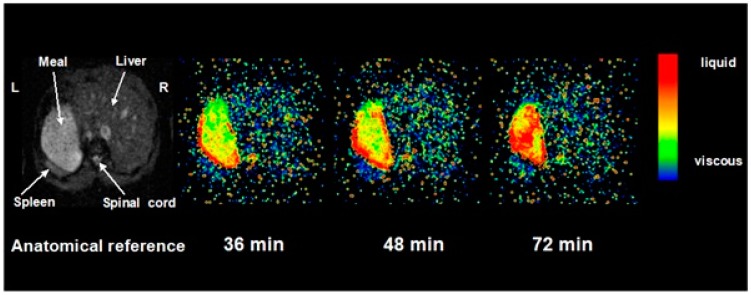
The time constant of the decay of image intensity (T2) maps of the stomach after ingestion of locust bean gum with viscosity of 17 Pa·s. Reproduced from [11], American Physiological Society 2001, with permission.

**Figure 4 nutrients-11-01147-f004:**
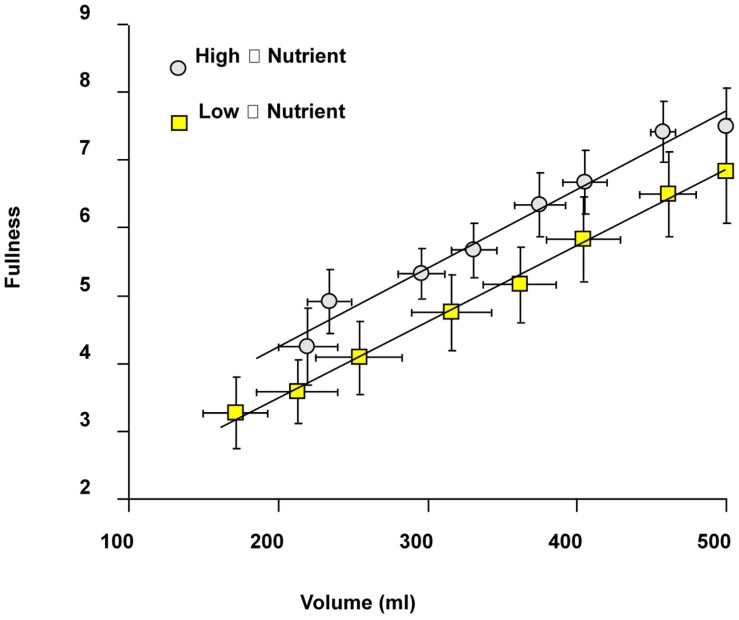
Correlation between sense of fullness and gastric volume after a high viscosity (17 Pa·s) and a low viscosity (0.01 Pa·s) nutrient meal containing 1350 kJ made up of 37% fat and 63% carbohydrate. The high viscosity meal generated a greater sensation of fullness. Reproduced from [11], American Physiological Society 2001, with permission.

**Figure 5 nutrients-11-01147-f005:**
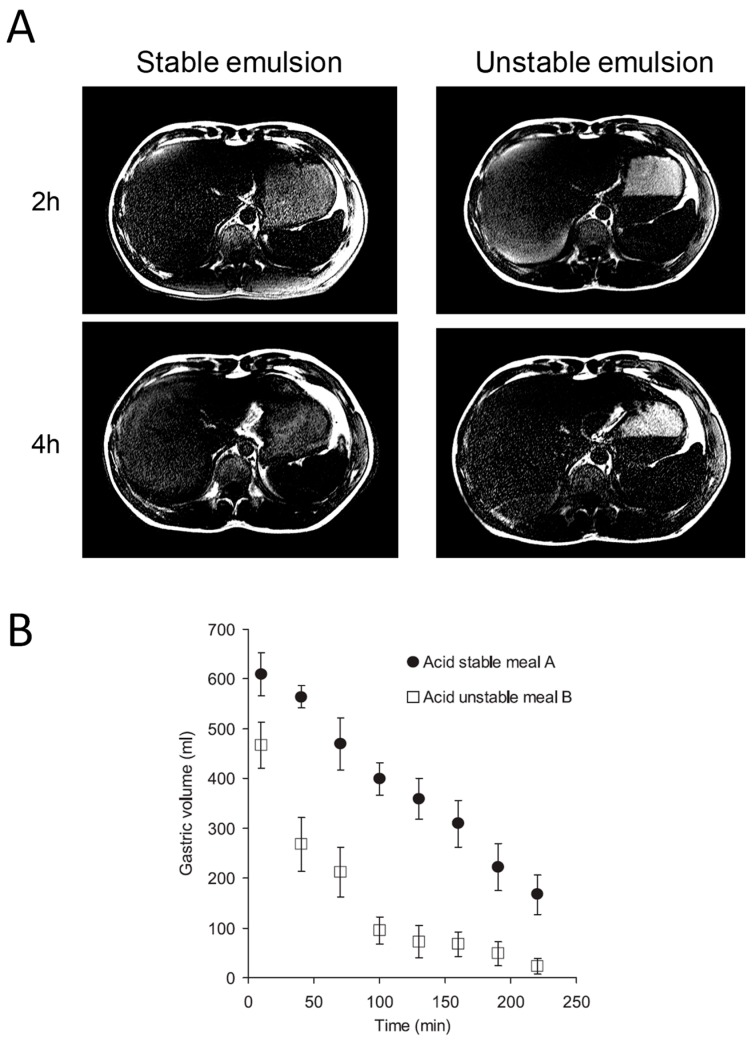
(**A**) MRI of isocaloric stable and unstable emulsions imaged 2 and 4 h after ingestion. The unstable emulsion shows fat layering above the fluid phase, most of which has emptied by 4 h, while the stable emulsion remains homogeneous with slower overall emptying. (**B**) Gastric volume over time post ingestion showing faster emptying of the acid unstable meal. Reproduced from [20], American Physiological Society 2007, with permission.

**Figure 6 nutrients-11-01147-f006:**
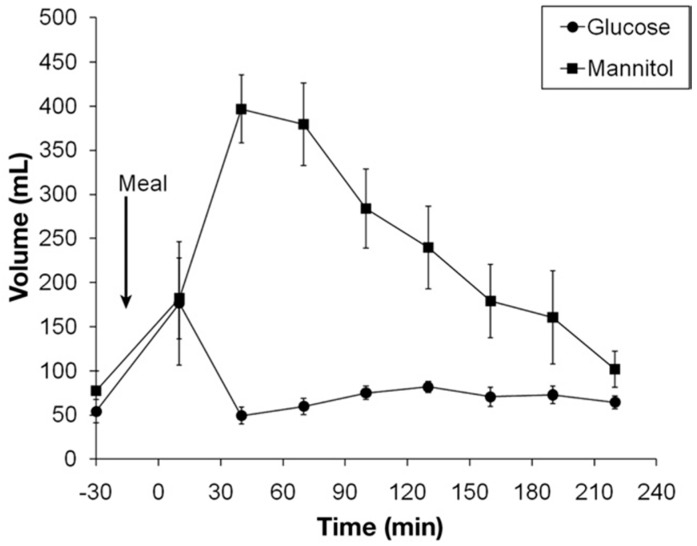
Postprandial time profile of small bowel water content (SBWC) in mL after ingestion of a 350 mL 5% glucose or mannitol test meal. While the glucose solution is absorbed almost as soon as it leaves the stomach the osmotically active but unabsorbable mannitol causes marked water secretion quadrupling the SBWC. Reproduced from [23], Elsevier 2010, with permission.

**Figure 7 nutrients-11-01147-f007:**
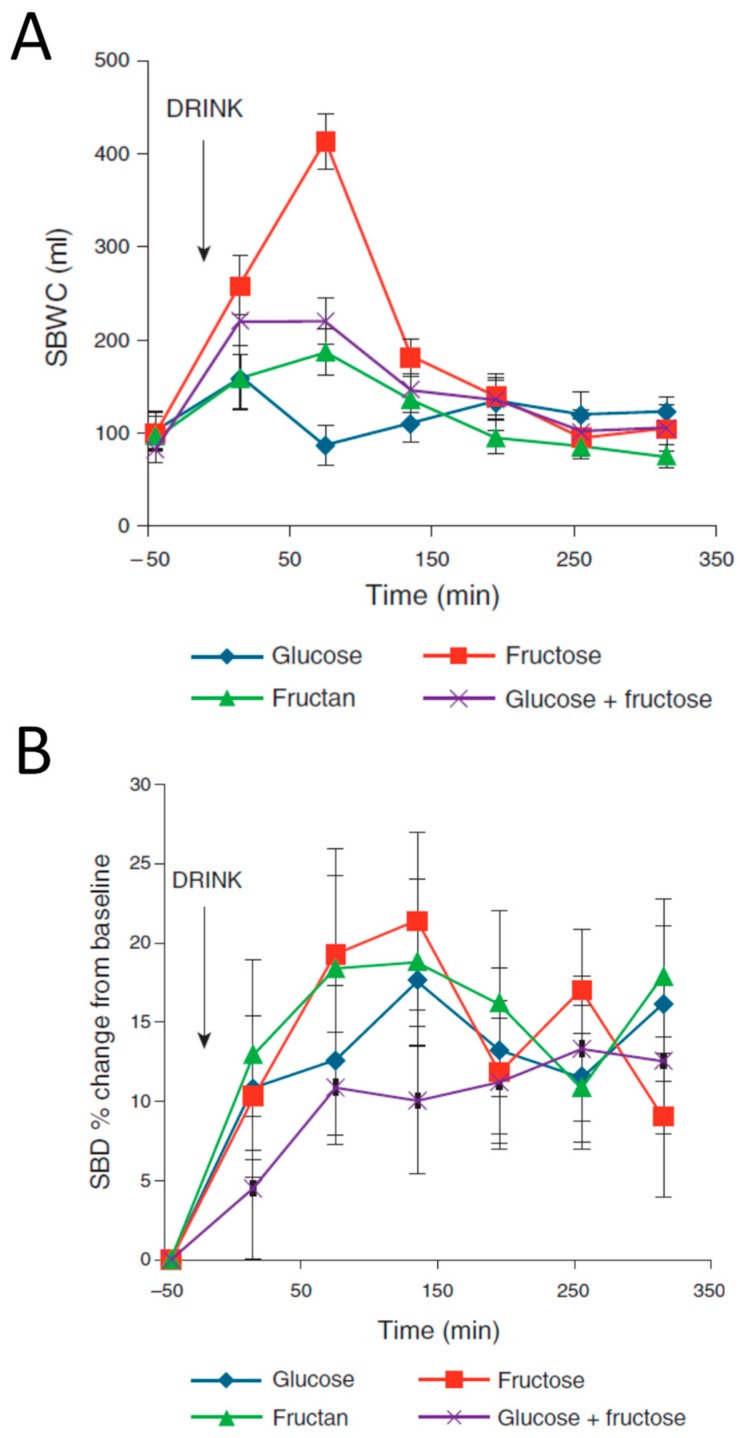
(**A**) Small bowel water content (SBWC) time profile in response to glucose, fructose, fructan and combined fructose drinks. (**B**) Small bowel diameters (SBD) in response to these interventions. Reproduced from [30], Wolters Kluwer Health 2014, with permission.

**Figure 8 nutrients-11-01147-f008:**
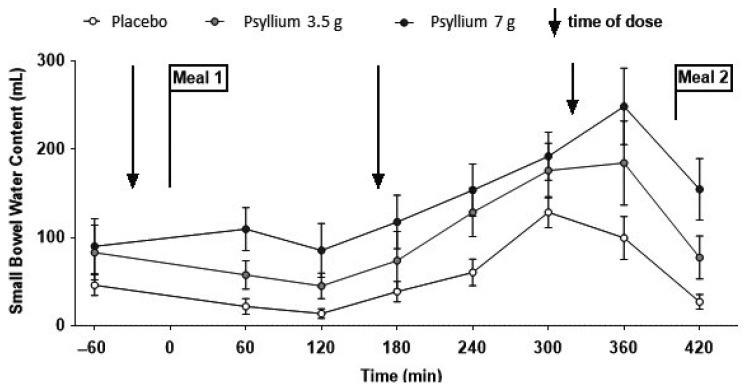
Small bowel water content during the study day during a RCT of psyllium 3.5 g and 7 g versus maxtodextrin 14 g three times daily (shown by arrows), showing a rise in SBWC particularly after the second and third dose. Reproduced from [32], John Wiley and Sons 2019, with permission.

**Figure 9 nutrients-11-01147-f009:**
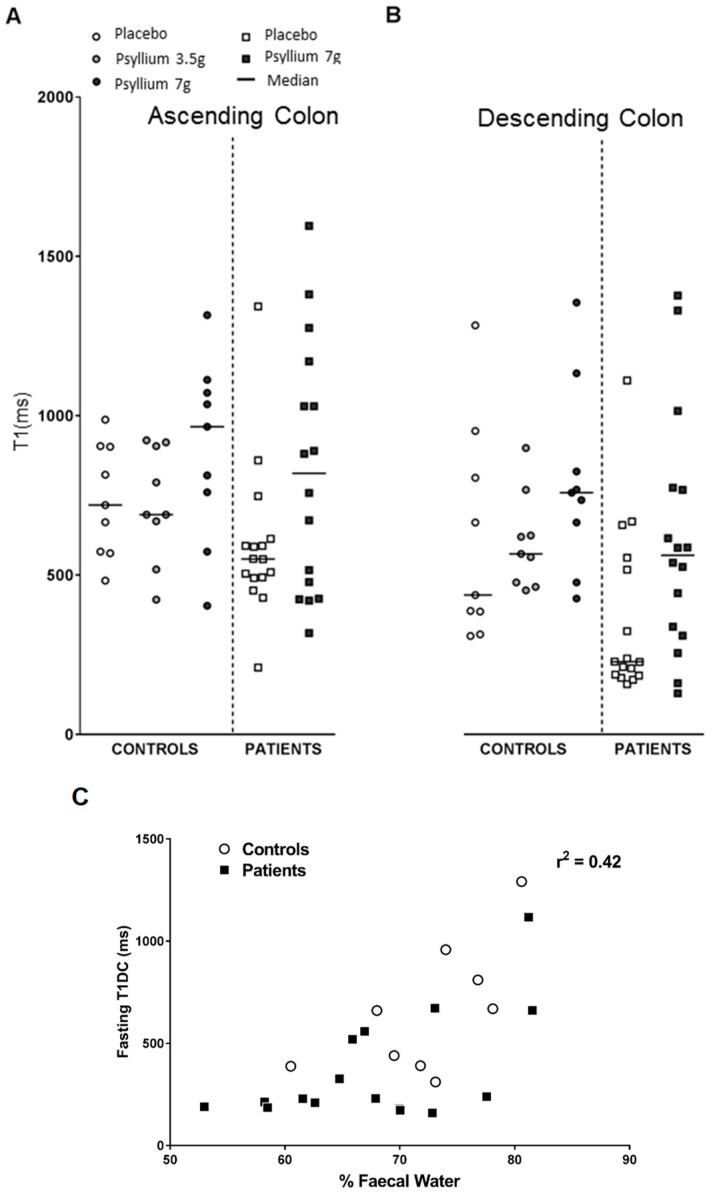
(**A**) Recovery of the longitudinal magnetization after a radiofrequency excitation (T1) values in ascending and (**B**) descending colons of healthy volunteers and constipated patients given psyllium 3.5 g or 7 g t.d.s. (**C**) Fasting T1 in descending colon and its correlation with % water as assessed by freeze drying of the next stool passed after the MRI scan. Reproduced from [32], John Wiley and Sons 2019, with permission.

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
