# Peer review of "Intraluminal Impact of Food: New Insights from MRI"

_nutrients, 2019, doi:10.3390/nu11051147_

Round 1

Reviewer 1 Report

The authors present a narrative review about the merits of MRI for studying the transit of food through the GI tract. It reads very nicely but there is quite some room for improvement. Specifically:

- The abstract appears incomplete. It should list ''results'' but also end with a conclusion

- The paper is poorly referenced. In several places/paragraphs there are no references.

- Although I would be the first to agree that this group is world-leading in this field I am disappointed that no studies from other groups are mentioned. In fact, the paper is basically a haphazard set of examples rather than a more or less complete overview of what can be done with MRI in this domain. I think that is a pity because the authors appear well positioned to write an authorative review. This piece reads like a promotional leaflet, that merely hghlights examples from one group. Thus, I would encourage the authors to be more clear about the scope and purpose of the review, to be more complete, also referencing some of the other work done e.g. by Steingoetter et al.

- Also, the general tone is very narrative and there is not much critical review, nor are limitations or pitfalls of the technique discussed.

- The title mentions ''new insights" but the piece is rather an overview of what has been done so far.

-Reusing figures from previous papers is generally fine, but may need formal approval and certainly appropriate referencing. 

-It would be good to have a brief introductory explanation about how MRI works and generates different types of images.

-I would appreciate somewhat of an outlook on where we could or should go from here. Are the new developments (MRI develops quite fast) that can be applied in this domain?

- Section 11 needs a better title, is very brief, and lacks references

-Section 12 is more of a summary and ends a bit suddenly.

Minor:

-I am not sure a PhD thesis can be referenced in this way; is it publicly accessible? Please provide a link.

-Figure 6 y-axis would benefit from a label ''SBWC" similar to Fig 7

Author Response

Reviewer 1

Thanks for your time and comments Our responses are listed below

The authors present a narrative review about the merits of MRI for studying the transit of food through the GI tract. It reads very nicely but there is quite some room for improvement. Specifically:

- The abstract appears incomplete. It should list ''results'' but also end with a conclusion

Response: Apologies This was an error. What is listed under “abstract” is in fact the introductory paragraph

We have now created an abstract

- The paper is poorly referenced. In several places/paragraphs there are no references

Response: We agree and have increased the references substantially

- Although I would be the first to agree that this group is world-leading in this field I am disappointed that no studies from other groups are mentioned. In fact, the paper is basically a haphazard set of examples rather than a more or less complete overview of what can be done with MRI in this domain. I think that is a pity because the authors appear well positioned to write an authorative review. This piece reads like a promotional leaflet, that merely hghlights examples from one group. Thus, I would encourage the authors to be more clear about the scope and purpose of the review, to be more complete, also referencing some of the other work done e.g. by Steingoetter et al.

Response: We agree that we have been rather parochial. Initially we were almost the only group doing such work but that has changed significantly. We have now added a number of sections highlighted in yellow to emphasize the important work being done in the area by others with 13 new references..

- Also, the general tone is very narrative and there is not much critical review, nor are limitations or pitfalls of the technique discussed.

Response:

While we agree with this comment we don’t think detailed analysis of the complexities of MRI  would be useful for the readers of Nutrients. We have added several comments about limitations in a modified section 12 entitled “Limitations”.

- The title mentions ''new insights" but the piece is rather an overview of what has been done so far.

Response ~Our revised conclusions do now point towards the future possibilities which we hope will stimulate others

-Reusing figures from previous papers is generally fine, but may need formal approval and certainly appropriate referencing. 

Response: all figure reproduction permission have been obtained

-It would be good to have a brief introductory explanation about how MRI works and generates different types of images.

Response: We have now added a new section 1 covering this as requested

-I would appreciate somewhat of an outlook on where we could or should go from here. Are the new developments (MRI develops quite fast) that can be applied in this domain?

We have now added this to section 13  on future developments

- Section 11 needs a better title, is very brief, and lacks references

Response: This is now section 12  entitled ”Limitations and how these might be overcome”

-Section 12 is more of a summary and ends a bit suddenly.

Response: We now have the section on future developments as requested.

Minor:

-I am not sure a PhD thesis can be referenced in this way; is it publicly accessible? Please provide a link.

Response: Unfortunately  we now find that this thesis is currently embargoed so it cannot be referenced but is listed as manuscript in preparation

-Figure 6 y-axis would benefit from a label ''SBWC" similar to Fig 7

Response: Figure 6 is reproduced as in the original publication

Reviewer 2 Report

This is a very well written review of exciting new use of MRI technology that can enhance our at times limited understanding of gastrointestinal physiology (motility and visceral sensation).

The only aspects of this review I felt to be missing were a discussion of the practical limitations of this technology - that is do the authors see a role for clinical or only research applications - and a further discussion of potential clinical or research applications beyond those mentioned.

Congratulations I enjoyed reading this contribution very much.

Author Response

Reviewer 2

This is a very well written review of exciting new use of MRI technology that can enhance our at times limited understanding of gastrointestinal physiology (motility and visceral sensation).

The only aspects of this review I felt to be missing were a discussion of the practical limitations of this technology - that is do the authors see a role for clinical or only research applications - and a further discussion of potential clinical or research applications beyond those mentioned.

Response: Thank you for your kind comments. We now include a limitations section  and have also added a comment about clinical applications at the end.

“ Some applications may well move MRI from its current purely research role into clinical practice.  Imaging the response of constipated patients to an osmotic laxative clearly distinguishes those with an inert hypomotile colon from those with a hyperactive irritable bowel and may well guide treatment in the future 44;45. Whether individual differences in responses to dietary items might also guide diet therapy remains to be determined.”

Reviewer 3 Report

This is a new and interesting paper. Hovewer, this is a review and there is an important methodology lack: how did the  authors find the papers they reviewed? In which  database did they performe their search? which "Mesh headings" did they use? Did they use "Boolean operators" ("and"/"or")? Which were the "Search limits" (interval time, human studies., language etc)? All this elements must be specified.

Author Response

Reviewer3

This is a new and interesting paper. Hovewer, this is a review and there is an important methodology lack: how did the  authors find the papers they reviewed? In which  database did they performe their search? which "Mesh headings" did they use? Did they use "Boolean operators" ("and"/"or")? Which were the "Search limits" (interval time, human studies., language etc)? All this elements must be specified.

Response: Thank you for your kind comments. We should make clear this was not commissioned as a systematic review and is not intended to be such . It is a personal perspective and as such description of search technique is not we feel appropriate.

We now make this clear in the introductory paragraph

“This selected review will use studies mostly from our own laboratory but also from others to describe how food alters intraluminal content in the stomach, small and large intestine

Round 2

Reviewer 3 Report

It can be published